# Increased phagocytosis in the presence of enhanced M2-like macrophage responses correlates with increased primary and latent HSV-1 infection

Ujjaldeep Jaggi[1], Mingjie Yang[2], Harry H. Matundan[1], Satoshi Hirose[1], Prediman K. Shah[2], Behrooz G. Sharifi[2], Homayon Ghiasi[1]*

1 Center for Neurobiology and Vaccine Development, Ophthalmology Research, Department of Surgery, Cedars-Sinai Burns & Allen Research Institute, Los Angeles, CA, United States of America, 2 Oppenheimer Atherosclerosis Research Center, Cedars-Sinai Smidt Heart Institute, and Department of Surgery, Los Angeles, CA United States of America

☯ These authors contributed equally to this work.
* ghiasih@CSHS.org

**Data Availability Statement:** All relevant data are within the manuscript and its Supporting Information files.

## Abstract

After HSV-1 infection, macrophages infiltrate early into the cornea, where they play an important role in HSV-1 infection. Macrophages are divided into M1 or M2 groups based on their activation. M1 macrophages are pro-inflammatory, while M2 macrophages are anti-inflammatory. Macrophage phenotypes can shift between M1 or M2 *in vitro* and *in vivo* following treatment with specific cytokines. In this study we looked at the effect of M2 macrophages on HSV-1 infectivity using mice either lacking M2 (M2[-/-]) or overexpressing M2 (M2-OE) macrophages. While presence or absence of M2 macrophages had no effect on eye disease, we found that over expression of M2 macrophages was associated with increased phagocytosis, increased primary virus replication, increased latency, and increased expression of pro- and anti-inflammatory cytokines. In contrast, in mice lacking M2 macrophages following infection phagocytosis, replication, latency, and cytokine expression were similar to wild type mice. Our results suggest that enhanced M2 responses lead to higher phagocytosis, which affected both primary and latent infection but not reactivation.

## Author summary

Macrophages are one of the major infiltrates into the cornea of ocularly infected mice and may contribute to both disease and protection. Similar to $T_H1/T_H2$ T cells, macrophages also are divided into M1/M2 subtypes. However, very little is known about the role, if any, that M1/M2 macrophages play in HSV-1 primary and latent infection. Here we investigated the role of M1/M2 macrophages in HSV-1 latency-reactivation by generating M2[-/-] mice as well as conditional transgenic mice that overexpress M2 macrophages. We found that higher expression of M2 macrophages correlated with higher phagocytosis, higher primary virus replication and latency, and higher IL-4 and IFN-γ expression, while the

  

**Funding:** This study was supported by Public Health Service NIH grants RO1EY024649, RO1EY029160 and RO1EY029677 to HG. The funders had no role in study design, data collection and analysis, decision to publish, or preparation of the manuscript.

**Competing interests:** The authors have declared that no competing interests exist.

absence of M2 macrophages did not significantly alter HSV-1 infectivity compared with wild type mice. Thus, maintaining a balanced proportion of M1/M2 macrophage phenotype may be an effective way to control both primary and latent infection.

## Introduction

In immunopathological diseases like herpes stromal keratitis (HSK), caused by HSV-1 (Herpes Simplex Virus-1), both T cells and innate immune cells play a major role in combating the virus replication which could eventually lead to HSK if not properly managed [1, 2]. Macrophages are an important first line of defense against various pathogens and have both protective and pathogenic roles in a wide variety of autoimmune and inflammatory diseases [3]. As professional phagocytes, macrophages have multiple functions including cytokine/chemokine secretion, regulating the tumor environment, and phagocytosis. Additionally, macrophages attract and activate other cells of the adaptive immune system, especially T cells, to the site of infection [4]. Despite being the dominant infiltrate into the eye following ocular infection with HSV-1 [5], the role of macrophages in corneal disease is controversial [6–9].

Macrophages can polarize toward a classical, pro-inflammatory phenotype, termed M1, or an alternative anti-inflammatory phenotype, termed M2. These two subtypes differ in the cytokines and chemokines they secrete. Several studies have suggested that M1 macrophages, which express NOS, can enhance inflammation by secreting pro-inflammatory cytokines [10, 11]. Conversely, M2 macrophages express arginase (Arg1) and are thought to be anti-inflammatory immune cells that help control inflammation [10–14]. M2 macrophages also promote the differentiation of $T_H2$ cells and regulatory T cells, which play important roles in suppressing inflammation and enhancing tissue repair, while dampening $T_H1$ release of pro-inflammatory cytokines [12]. Activation of M2 macrophages depends on cytokines IL-4 and IL-13, which are required for $T_H2$ cell differentiation [12–14]. Both M1 and M2 macrophages are generated from untreated macrophages (M0) [15].

Little is known about the M1 and M2 macrophage sub-populations and their roles in disease. In the case of human immunodeficiency virus (HIV) infection, stimulating primary human-monocyte derived macrophages with IFN-γ and TNF-α (M1 polarization) remarkably reduced virus replication compared to human-monocyte derived macrophages without stimulation [16]. M2 activation has been shown to be beneficial in an RSV-induced bronchiolitis mouse model [17, 18]. However, a study in an MS model suggested that maintaining a proper balance between M1 and M2 populations was important [12].

We recently showed that after injecting colony-stimulating factor 1 (CSF-1), macrophages became polarized toward the M2 phenotype with more protection against primary and latent HSV-1 infection in the eye than when macrophages were polarized toward the M1 phenotype after injecting gamma interferon (IFN-γ) [19]. We also demonstrated that recombinant HSV-1 expressing IL-4 (an inducer of M2 responses) provided better protection than a recombinant HSV-1 expressing IFN-γ (an inducer of M1 responses) or wild type (WT) virus against virus replication, latency, reactivation, and eye disease in wild type mice [20]. To further study the potential role of M2 macrophages to protect against HSV-1 ocular infections, we generated novel *in vivo* models of M2 polarized macrophages. GATA3 is one of the transcription factors involved in M2 polarization [21] and evolutionarily highly conserved [22, 23]. GATA3 is required for $T_H2$ cell differentiation [24], generation of type 2 innate lymphoid cells [25], function of natural killer (NK) cells [26], and pathological humoral responses [27] and is highly relevant in models of inflammation, allergy, and tumorigenesis [28]. To investigate the role of

M2 polarization of macrophages in the HSV-1 ocular infection, we used myeloid-specific GATA3-defieicnt mice (designated as M2$^{-/-}$ mice hereafter) [29]. We have previously reported that the cardiac function is significantly improved in M2$^{-/-}$ mice compared with the WT in response to ischemic injury [29]. We have shown that this improvement was associated with the presence of large number of pro-inflammatory Ly6C$^{hi}$ monocytes/macrophages and few Ly6C$^{lo}$ reparative macrophages in the infracted myocardium of M2$^{-/-}$ mice compared with the WT mice. These results suggest that macrophage polarization, specifically GATA3 signaling play an important role in ischemic injury of myocardium. To further extend our investigation, we generated GATA3 overexpressed mice in a GATA3 conditional knockout mice background that we designated as M2-overexpressed (M2-OE) mice hereafter. M2-OE mice expressed human GATA3 and not mouse GATA3 in myeloid cells.

Using M2$^{-/-}$ (GATA3 deleted) and M2-OE (GATA3 overexpressed) mice, here we report that M2$^{-/-}$ mice behave similar to WT mice in terms of eye disease, viral titers, latency-reactivation, and levels of T cell exhaustion. Interestingly, we found increased virus replication, viral load, latency, IL-4 and IFN-γ in M2-OE mice. Our results suggest that maintaining a proper balance of macrophage sub-populations is crucial for controlling HSV-1 ocular infections.

## Results

### Generation and genotyping of mice

M2$^{-/-}$ and M2-OE mice were generated as described in Materials and Methods. The presence of Cre, GATA3 loxp, and Rosa26 in the resulting M2-OE pups were verified by PCR (Fig 1). Pups 5 and 13 contained all three genes and were used to breed M2-OE mice.

### *In vitro* polarization of macrophages isolated from WT, M2$^{-/,}$ and M2-OE mice

Peritoneal macrophages (PM) were isolated from WT, M2$^{-/-}$, and M2-OE mice after zymosan treatment as described in Materials and Methods. PM were stimulated with IL-4 (M2 polarization, Fig 2A) or IFN-γ and LPS (M1 polarization, Fig 2B) as described [19, 20]. Cells were harvested and total RNA was isolated. Expression of ARG1 and NOS2 mRNA, markers of M1 and M2 macrophages, respectively, were determined by qRT-PCR (Fig 2). Levels of ARG1 mRNA

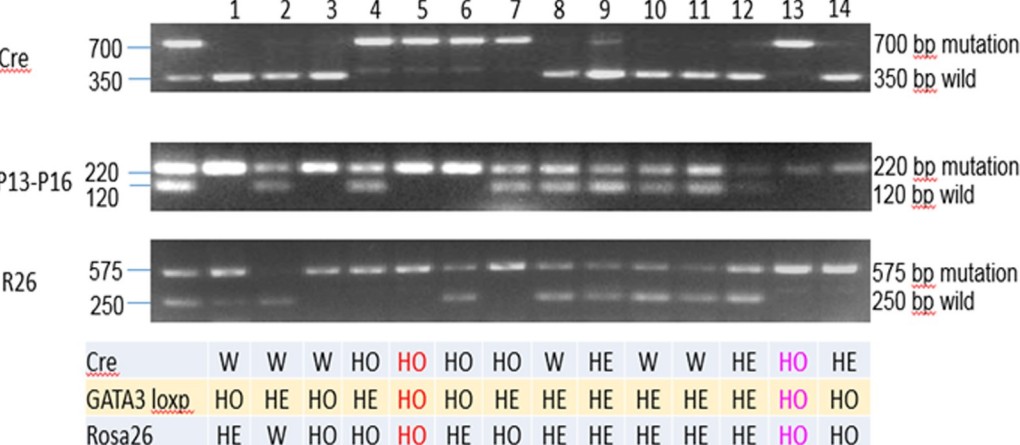

**Fig 1. Validation of M2-OE mice by PCR.** Presence of Cre, GATA3loxp (P13-16), and Rosa26 (R26) were verified by PCR as described in Materials and Methods. Two pups, pup #5 and pup #13 were homozygous (HO) for all three genes as compared to other pups which were heterozygous (HE). HO pups were thus used for subsequent breeding.

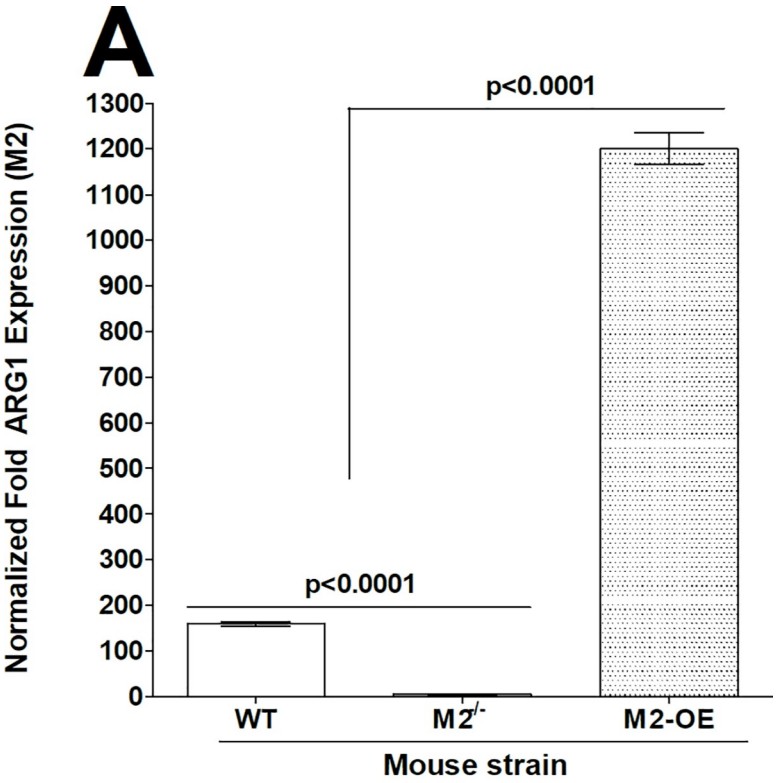

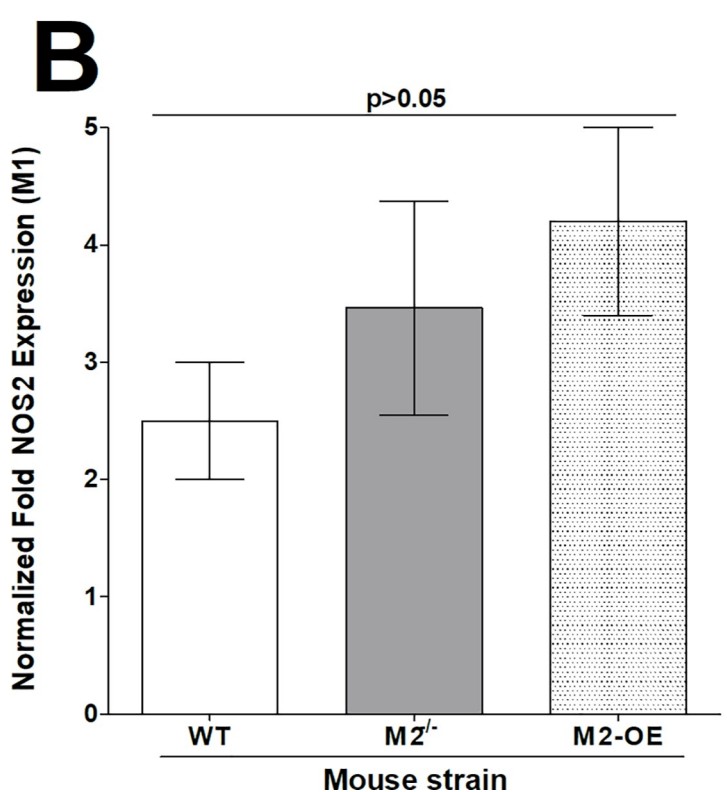

**Fig 2. Validation of macrophage phenotype *in vitro*.** Peritoneal macrophages (PM) from both male and female WT, M2$^{-/-}$, and M2-OE mice that had been treated with zymosan, were polarized into M1 and M2 phenotypes as described in Materials and Methods. Cells were then harvested; total RNA was isolated, and TaqMan RT-PCR was performed using ARG-1 (for M2 phenotype)- and NOS2 (for M1 phenotype)- specific primers. Expression of ARG1 (A) and NOS2 (B) mRNA was normalized to that of GAPDH RNA. Each bar represents the mean ± SEM from two independent experiments (N = 4).

were significantly higher in PM derived from M2-OE mice than in WT or M2$^{-/-}$ mice (Fig 2A, p<0.0001). Similarly, WT mice had higher levels of ARG1 mRNA than did M2$^{-/-}$ mice (Fig 2A, p<0.0001). No statistically significant differences were observed in NOS2 mRNA expression amongst the three mouse strains (Fig 2B, p>0.05). Together, these results establish that all three mouse strains can be polarized toward the M1 phenotype, but only WT and M2-OE mice can be polarized toward the M2 phenotype.

## Overexpression of M2 macrophages enhances phagocytic activity

Macrophages are professional phagocytes and antigen presenting cells [3], therefore, we measured phagocytosis in an *in vitro* assay using WT, M2$^{-/-}$, and M2-OE mice. Bone marrow (BM) derived macrophages were isolated from WT, M2$^{-/-}$, and M2-OE mice and infected with HSV-1 or mock infected and incubated with FITC labeled beads as described in Materials and Methods. The cells were then stained with F4/80 antibody and two hr after addition of beads the extent of phagocytosis was analyzed by flow cytometry. The number of FITC positive cells was higher in both mock and HSV-1 infected M2-OE BM cells (9.59% and 7.71%, respectively) than in WT and M2$^{-/-}$ PM in both mock and HSV-1 infected cells (Fig 3A). However, the level of FITC positive cells in WT and M2$^{-/-}$ cells with and without infection were similar (Fig 3A, WT, M2$^{-/-}$).

The average percentage of FITC positive cells from WT, M2$^{-/-}$, and M2-OE PM cells with and without infection was determined from three separate experiments. Percent phagocytosis in mock and HSV-1 infected M2-OE PM cells was significantly higher than in WT and M2$^{-/-}$ BM with and without infection (Fig 3B, M2-OE versus other groups, p<0.0001). In contrast, phagocytosis levels in WT BM was similar to that of M2$^{-/-}$ BM in mock and HSV-1 infected groups (Fig 3B, p>0.05). These results suggest that phagocytosis activity is enhanced in M2-OE mice independent of infection.

## Macrophage polarization affects HSV-1 replication during early infection *in vitro*

We have shown that polarization of macrophages from C57BL/6 mice toward the M1 phenotype resulted in lower virus replication when compared to M2 polarized macrophages [19]. To compare the effect of polarized macrophages on virus replication in M2$^{-/-}$, M2-OE, and WT mice, macrophages were isolated from these mouse strains and either left untreated or polarized using IFN-γ and LPS or IL-4 as we described previously [19]. We found no difference in virus titers between unpolarized cells from the three mouse strains (Fig 4A). Polarization toward M2 by IL-4 stimulation had no impact on virus replication in M2$^{-/-}$ or M2-OE macrophage populations, while IL-4 stimulation in WT macrophages elevated virus titers at 12 hr post infection (PI) however, titers declined thereafter (Fig 4B). Polarization toward M1 by IFN-γ stimulation increased viral titers in macrophages from M2-OE mice over that seen in macrophages from WT or M2$^{-/-}$ mice at 12 hr but declined to levels similar to WT and M2$^{-/-}$ by 24 hr PI (Fig 4C). These results show that although viral replication is initially higher in M2 polarized WT macrophages and M1 polarized macrophages from M2-OE mice, macrophages

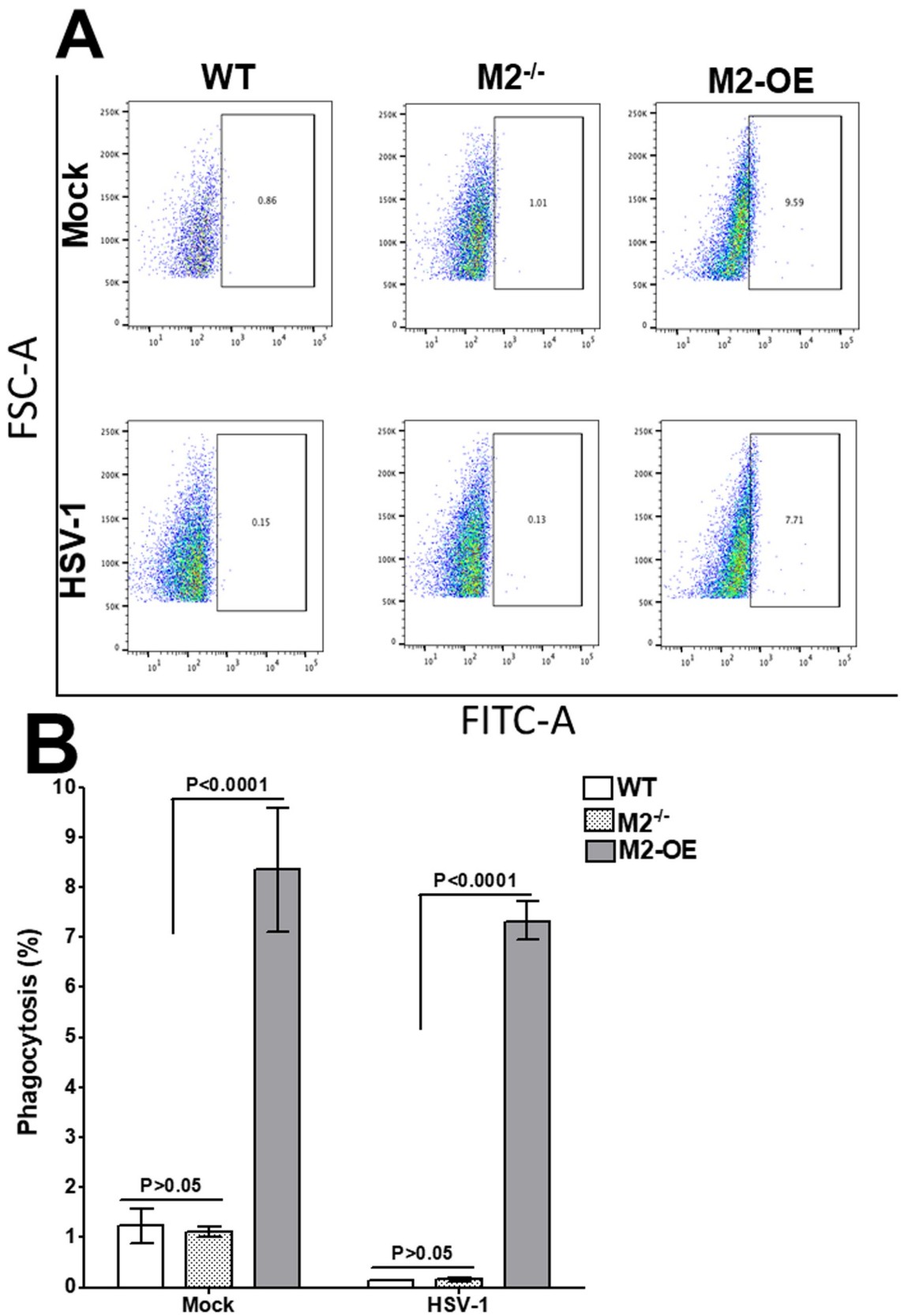

**Fig 3. Phagocytosis assay on *in vitro* derived bone marrow-derived (BM) macrophages.** Cells from BM-derived WT, M2$^{-/-}$, and M2-OE mice were cultured, differentiated into macrophages, and cultured in 24-well plates as described in Materials and Methods. After overnight culture, adhered cells were infected with HSV-1 and incubated with latex beads-rabbit IgG-FITC complex. Cells were stained with F4/80 AF 564 antibody and subjected to flow cytometry analysis. Panel A) representative plots of WT, M2$^{-/-}$, and M2-OE macrophages; and B) Percentage phagocytosis plots from three separate experiments.

can limit virus growth independent of polarization status as we reported for macrophages from WT mice [30].

## HSV-1 replication is increased in M2-OE mouse eyes

Previously we have shown that macrophages from WT mice after polarization towards M1 state had lower viral replication in comparison to macrophages polarized towards M2 state [19, 20]. To determine the effects of M2 macrophages on HSV-1 replication *in vivo*, WT, M2$^{-/-}$, and M2-OE mice were ocularly infected with 2 X 10$^5$ PFU/eye of virulent HSV-1 strain McKrae. Tear films were collected from day 1 to day 7 PI from ten mice per group and virus titers were determined by standard plaque assays as we described in Materials and Methods. Virus titers were similar in WT and M2$^{-/-}$ mice (Fig 5A, p>0.05). M2-OE mice had significantly higher viral titers on days 2, 3, and 4 PI in comparison to WT and M2$^{-/-}$ mice (Fig 5A, p<0.05).

Virus titration in the eyes of infected mice described above (Fig 5A), showed peak virus replication on day 4 PI. To confirm the above results, additional groups of WT, M2$^{-/-}$, and M2-OE mice were infected as above. Corneas from these infected mice were isolated on day 4 PI and total RNA was isolated and subjected to TaqMan RT-PCR to estimate levels of gB mRNA using GAPDH mRNA levels in each sample as an internal control. The results show a significantly higher gB copy number in M2-OE mice than in WT and M2$^{-/-}$ mice (Fig 5B, p<0.001). As expected, no significant differences were detected between WT and M2$^{-/-}$ mice groups (Fig 5B, p>0.05). These results suggest that overexpression of M2 macrophages enhances viral replication and viral load in infected mice, while the absence of M2 in M2$^{-/-}$ mice did not affect virus replication when compared with WT mice. Our current study supports our previous studies with regards to higher virus titer in M2-OE mice as compared to WT or M2$^{-/-}$ mice [19, 20].

## IL-4 and IFN-γ transcripts are elevated in HSV-1 infected corneas of M2-OE mice

To understand the cause of higher viral replication in the eye of M2-OE infected mice compared with WT and M2$^{-/-}$ mouse groups (Fig 5, above), we sought to determine expression levels of the pro-inflammatory cytokine IFN-γ and anti-inflammatory cytokine IL-4 in the corneas of infected mice. WT, M2$^{-/-}$, and M2-OE mice were ocularly infected with 2 X 10$^5$ PFU/eye of virulent HSV-1 strain McKrae as above. Mock infected mice in each group were used to normalize expression of each transcript in corneas of the three infected mouse strains. On day 4 PI, corneas from each mouse were pooled, total RNA was extracted, and expression of IFN-γ and IL-4 transcripts were measured by qRT-PCR. IFN-γ and IL-4 expression in mock infected mice from each group was used as a baseline control to estimate their relative transcript levels in corneas of infected mice. GAPDH expression was used to normalize the relative expression of each transcript. IFN-γ levels were significantly higher in M2-OE infected mice than in WT and M2$^{-/-}$ mice (Fig 6A, p<0.0001), while no differences were observed in IFN-γ levels in WT and M2$^{-/-}$ mice (Fig 6A, p>0.05). Similar to IFN-γ, IL-4 levels were significantly higher in M2-OE mice than in WT and M2$^{-/-}$ mice (Fig 6B, p<0.01), while no

## A (Mock)

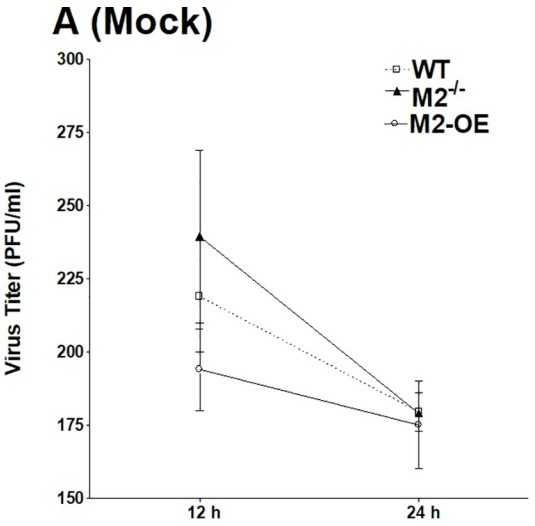

## B (+IL-4)

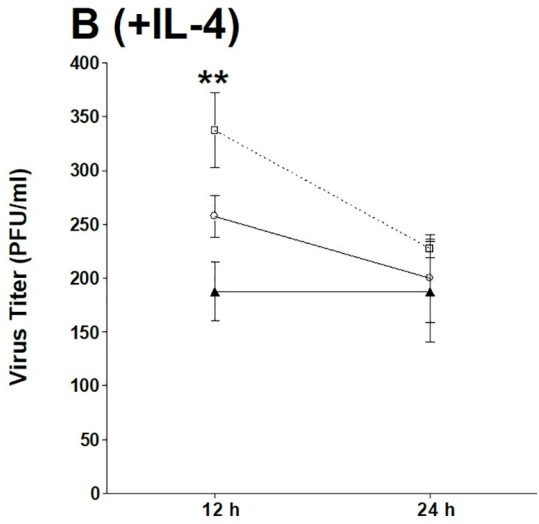

## C (+IFNγ)

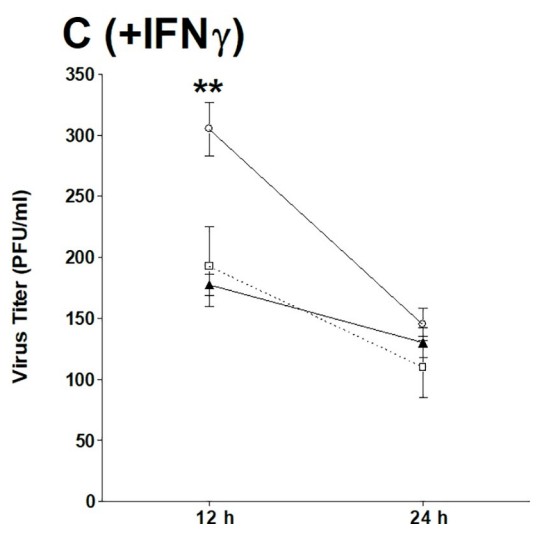

**Fig 4. Effect of macrophage polarization on HSV-1 replication *in vitro*.** WT, M2$^{-/-}$, and M2-OE mice were treated with zymosan and PM from WT, M2$^{-/-}$, and M2-OE mice were harvested. Isolated cells were mock treated for 24 hr (A), incubated for 24 hr with either IL-4 to generate M2 macrophages (B) or IFN-γ and LPS to generate M1 macrophages (C). Cells were then infected with 10 PFU/cell of McKrae virus for 1 hr. Infected cells were harvested at 12 and 24 hr PI and virus titers were determined using a standard plaque assay on RS cells. Each point represents the mean ± SEM from two independent experiments (N = 4).

differences were observed in IL-4 levels in WT and M2$^{-/-}$ mice (Fig 6B, p>0.05). These results suggest that higher expression of M2 macrophages in M2-OE mice than in M2$^{-/-}$ and WT mice leads to elevated expression of both pro-inflammatory and anti-inflammatory cytokines.

IFN-γ and IL-4 mRNA expression in naive mice for each group was used as a baseline control to estimate relative expression of each transcript in corneas of infected mice. Expression of each gene was normalized to GAPDH mRNA. Each point represents the mean ± SEM from two independent experiments (N = 3). Panels: A) IFN-γ mRNA expression; and B) IL-4 mRNA expression.

## Role of M2 macrophages on survival and eye disease in infected mice

Survival over 4 wk was monitored in two separate experiments using groups of 20 WT, M2$^{-/-}$, or M2-OE mice that had been infected ocularly in both eyes with 2 X 10$^5$ PFU/eye of McKrae virus. All 20 infected mice in the WT, M2$^{-/-}$, and M2-OE groups survived ocular infection (p = 1; ANOVA). These results suggest that neither the absence, nor overexpression, of M2 alters survival in ocularly infected mice compared with WT mice.

To determine the effect of M2 overexpression or absence on corneal scarring and angiogenesis, WT, M2$^{-/-}$, and M2-OE mice were infected with HSV-1 McKrae. Surviving mice were scored for corneal scarring and angiogenesis on days 2, 14, 21, and 28 PI as described in Materials and Methods. No statistically significant differences in corneal scarring (Fig 7A, p>0.05) or angiogenesis (Fig 7B, p>0.05) were observed between the three mouse groups. These results suggest that viral replication in this model of ocular HSV-1 infection does not lead to the severity of eye disease in M2-OE mice and that the presence or absence of M2 macrophages does not play a role in eye disease in these mice.

## Overexpression of M2 macrophages increases latency but not reactivation compared with WT and M2$^{-/-}$ mice

Latency associated transcript (LAT) is the only viral transcript that is abundantly expressed in TG of latency infected mice and increases viral latency [31–33]. The effect of M2 on the ability of LAT to increase latency was investigated in WT, M2$^{-/-}$, and M2-OE mice. Replication of HSV-1 in eyes during the first four days of infection was higher in M2-OE mice than in WT and M2$^{-/-}$ mice (Fig 5). The eye swab and gB transcript data showed a significant increase in virus replication in M2-OE mice, suggesting that M2 overexpression may enhance acute HSV-1 infection and higher virus replication may increase latency-reactivation in infected mice. TG from surviving WT, M2$^{-/-}$, and M2-OE mice infected ocularly with HSV-1 were harvested on day 28 PI and latency levels were determined by qRT-PCR based on HSV-1 LAT expression as described in the Materials and Methods. Consistent with elevated viral replication, latency was enhanced in M2-OE mice suggesting that M2 overexpression influences the amount of latency that is established and/or maintained.

Previously we reported a lack of correlation between primary virus titer in the eye and TG with the level of viral DNA in latent TG and time to reactivation using WT mice [34]. To determine whether higher virus replication in the eye and higher latency correlate with faster

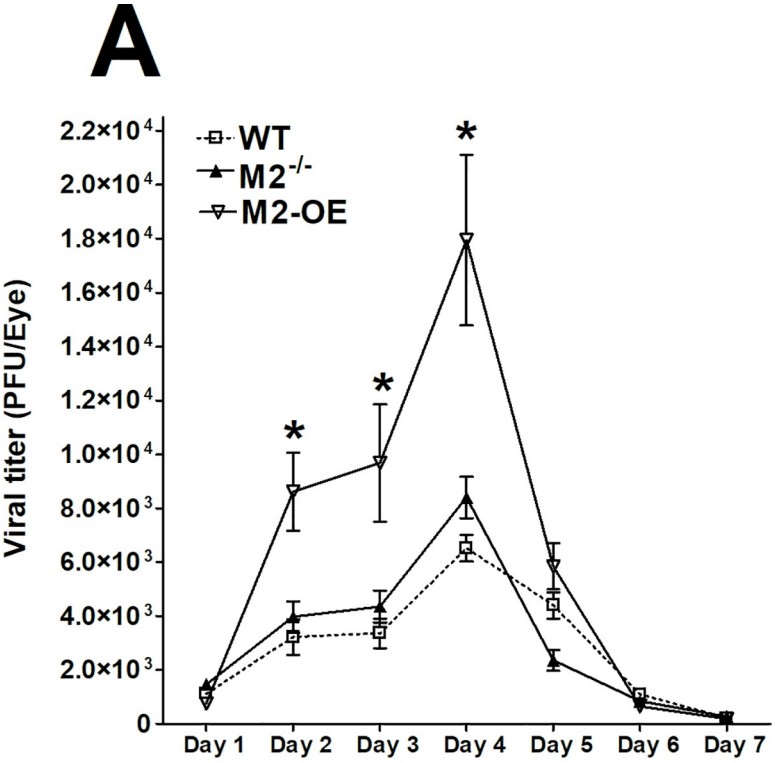

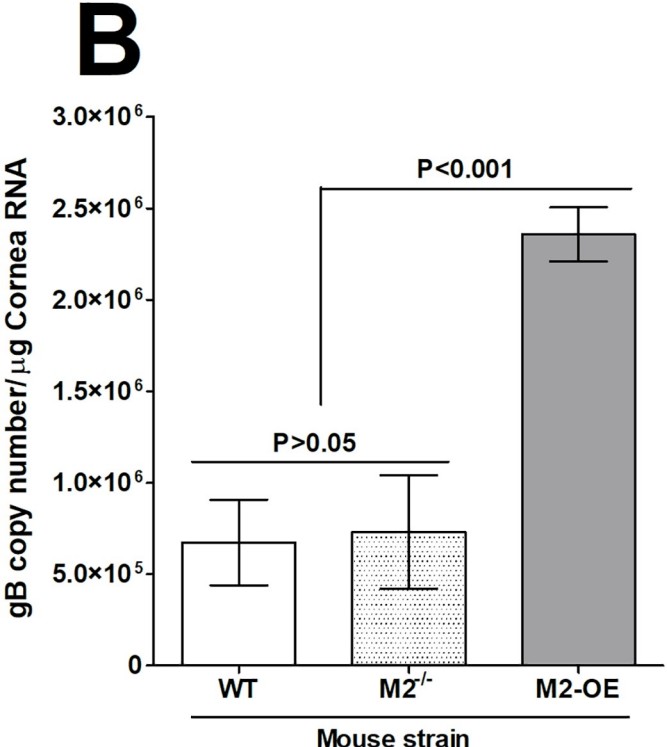

**Fig 5. Virus replication and gB expression in infected mice.** (A) Virus replication in tears of infected mice. WT, M2[-/-], and M2-OE mice were ocularly infected with 2 X 10[5] PFU/eye of McKrae virus without corneal scarification. Tear films were collected on days 1–7 PI and virus titers were determined by standard plaque assays as described in

Materials and Methods. Each point represents the mean ± SEM of 20 eyes per group. (B) Expression of gB in corneas of infected mice. WT, M2$^{-/-}$, and M2-OE mice were infected ocularly as above. gB expression in corneas of infected mice was determined on day 4 PI. In each experiment, the estimated relative copy number of HSV-1 gB was calculated using standard curves generated from pAC-gB1. Briefly, DNA template was serially diluted 10-fold such that 5 μl contained from $10^3$ to $10^{11}$ copies of LAT, then subjected to TaqMan RT-PCR with the same set of primers. By comparing the normalized threshold cycle of each sample to the threshold cycle of the standard, the copy number for each reaction was determined. GAPDH expression was used to normalize the relative expression of each transcript in corneas and TG of infected mice. Each bar represents the mean ± SEM from 3 mouse corneas. Panels: A) Virus titer; and B) gB expression.

reactivation in M2-OE mice we next tested explant reactivation from latency in WT, M2$^{-/-}$, and M2-OE mice ocularly infected with 2 X $10^5$ PFU/eye of HSV-1. Virus reactivation was analyzed by explanting individual TG from infected mice on day 28 PI as described in Materials and Methods. In contrast to the latency level, time of reactivation was similar between WT, M2$^{-/-}$, and M2-OE mice (Fig 8B. p>0.05). Thus, in M2-OE mice the time to explant reactivation did not correlate with the level of latency.

### Levels of CD4, CD8α, PD1, and Tim3 mRNAs in TG of latently infected M2$^{-/-}$, M2-OE, and WT mice

To investigate the effect of M2 macrophages on CD4 and CD8 expression as well as on PD1 and Tim3 exhaustion markers [32], relative levels of CD4, CD8, PD1, and Tim3 expression were determined by real-time PCR of total TG extracts. The results are presented as "fold" increase compared to baseline mRNA levels in TG from uninfected naive mice for each group (Fig 9). CD4 levels in TG of M2$^{-/-}$ and M2-OE mice were similar (Fig 9A, p>0.05) and both were significantly higher than CD4 levels in WT mice (Fig 9A, p<0.01). In contrast, CD8 transcript levels were significantly higher in M2$^{-/-}$ mice than in WT and M2-OE mice (Fig 9B, p<0.001), while CD8 mRNA levels were similar in WT and M2-OE mice (Fig 9B, p>0.05). No significant differences in PD1 expression was detected between WT and M2$^{-/-}$ mice (Fig 9C, p>0.05) and similarly, no differences were detected between M2$^{-/-}$ and M2-OE mice (Fig 9C, p>0.05). Finally, the level of PD1 expression was not significantly different between M2$^{-/-}$ and M2-OE mice (Fig 9C, p>0.05). Tim3 transcript levels were similar between M2$^{-/-}$ and M2-OE mice (Fig 9D, p>0.05) but both were significantly higher than in WT mice (Fig 9B, p<0.001). These findings suggest that knockout or overexpression of macrophages have no impact on CD4, CD8, PD1 or Tim3 transcripts levels.

## Discussion

We have previously shown that transient polarization of macrophages *in vivo* can alter the phenotype of macrophages in WT mice, thus affecting the level of latency-reactivation in ocularly infected mice [19, 20]. In the current study, rather than using CSF-1 or IL-4 and IFN-γ to push the phenotypes of WT mice toward M2 or M1, respectively, we generated knockout mice lacking M2 (M2$^{-/-}$) or conditional transgenic mice overexpressing M2 (M2-OE). The phenotypes of macrophages from M2$^{-/-}$ and M2-OE mice were validated *in vitro*. Macrophages from both M2$^{-/-}$ and M2-OE mice were polarized toward the M1 phenotype compared to WT macrophages, while only macrophages from WT and M2-OE mice were polarized toward the M2 phenotype. This is similar to our previous published studies using WT rather than knockout or conditional transgenic mice [19, 20].

Macrophages are considered professional phagocytes that are important for viral clearance. Our experiments demonstrate that M2-OE mice have significantly higher phagocytic activity than either WT or M2$^{-/-}$ mice. These results are consistent with reports showing that M2

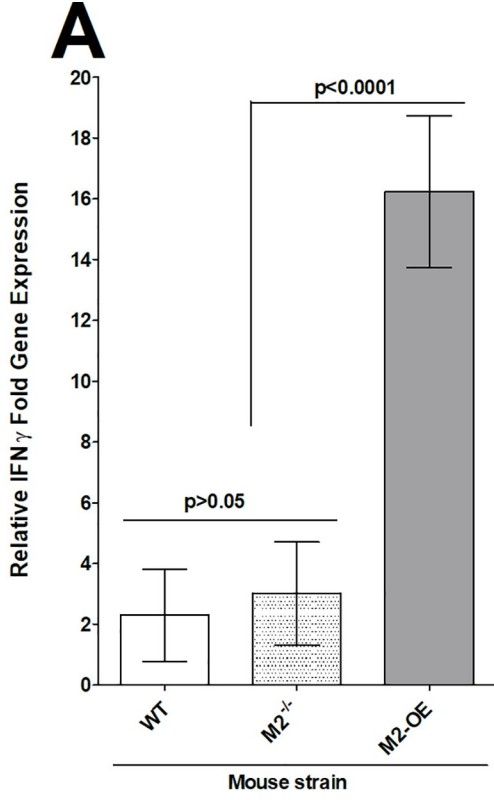

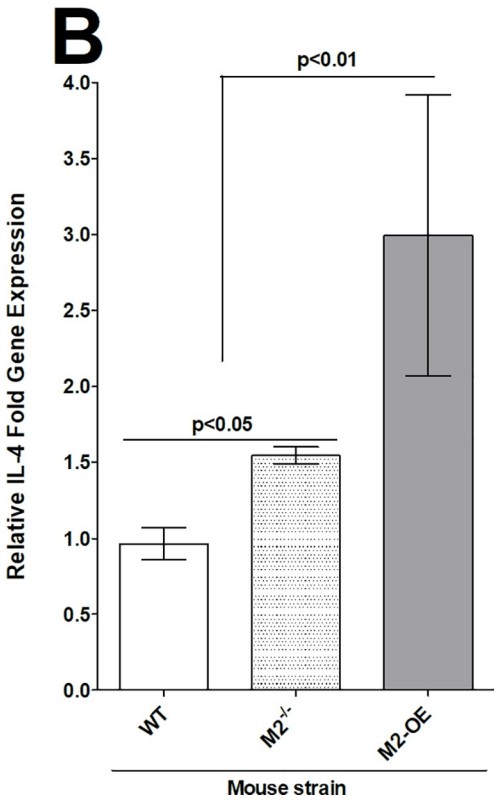

**Fig 6. Detection of IL-4 and IFN-γ in the corneas of infected mice.** WT, M2$^{-/-}$, and M2-OE mice were ocularly infected in both eyes with 2 X 10$^5$ PFU/eye of McKrae virus. On day 4 PI, corneas from each mouse were pooled and homogenized. Total RNA was extracted and subjected to qRT-PCR analysis.

macrophages have higher phagocytic activity than M1 macrophages [35, 36]. We found that HSV-1 infection did not alter higher phagocytosis in M2-OE mice. In contrast, hepatitis C virus (HCV) infection has been shown to inhibit phagocytosis activity of M1 and M2 macrophages [37]. The difference between these results is likely due to differences between HSV-1 and HCV viruses. In addition, in contrast to HSV-1 [19, 20], HCV inhibits monocyte differentiation to either M1 or M2 macrophages [37].

We previously showed that HSV-1 replication is impaired in macrophages from WT C57BL/6 mice polarized toward the M1 phenotype *in vitro* [19, 20]. Our current results in macrophages from WT C57BL/6 mice agree with these findings. Interestingly, we found that HSV-1 replication in the eye of ocularly infected M2-OE as well as HSV-1 gB transcript levels in corneas of infected mice were significantly higher than in WT or M2$^{-/-}$ mice. These results are striking as M2 macrophages are generally thought to have a protective role and we previously showed that polarization of mouse macrophages to the M2 phenotype *in vivo* reduced viral replication and latency in WT mice [19, 20]. These differences between the previous studies and our current results could not be due to the use of human GATA3 in place of mice GATA3 in M2-OE mice since GATA3 is highly conserved between mouse and human [22, 23]. Moreover, in our previous reports, ARG1 levels measured from peritoneal macrophages from WT mice and also RAW264.7 cell line increased approximately 2 to 4 fold in M2 polarized cells whereas, in our current study, the presence of enhanced M2 macrophages in M2-OE mice causes ARG1 levels to enhance by about 1200 fold as compared to WT or M2$^{-/-}$ mice. This suggests that excessive expression of M2 macrophages may have detrimental effects on virus replication and latency while a balanced ratio of M1/M2 macrophages is beneficial to host as shown in our previous studies [19, 20]. Also, it is possible that chronic imbalance in the macrophage population (i.e. increase or absence of M2 macrophages) in our current study can suppress the functional activity of M1 macrophages, causing an inability to clear the virus efficiently. However, as we described above, M2-OE macrophages display higher phagocytosis, which may contribute to increased virus replication. Similar to this study, replication of HIV type 1 has been shown to be enhanced after phagocytosis of apoptotic cells [38]. Furthermore, it has been reported that macrophage phagocytosis may contribute to increased foot-and-mouth disease virus infectivity [39]. In accordance with this, our study depicts another correlation between GATA3 expression and phagocytosis activity. Previously it was reported that microRNA-720 inhibits M2 polarization and phagocytosis by inhibiting GATA3 expression [40].

M2 macrophages are known to be involved in various physiologic and pathological processes including homeostasis, repair mechanisms, and are largely thought to serve an anti-inflammatory function [12–14]. M1 macrophages, on the other hand, are known to express pro-inflammatory cytokines and co-stimulatory molecules. Activation of M2 macrophages depends on cytokines IL-4 and IL-13, which are required for T$_H$2 cell differentiation [12–14]. Interestingly, we observed elevated levels of the pro-inflammatory cytokine IFN-γ as well as anti-inflammatory cytokine IL-4 in M2-OE mice. Elevated levels of IFN-γ and IL-4 correlated with higher viral replication and increased latency, but not reactivation.

We found that increased virus replication during primary infection in the eye of infected mice and elevated latency in TG of latently infected mice did not correlate with reactivation in M2-OE mice. This is consistent with our previous studies in which we showed that reactivation is independent of increased primary virus replication and viral load [34, 41–43].

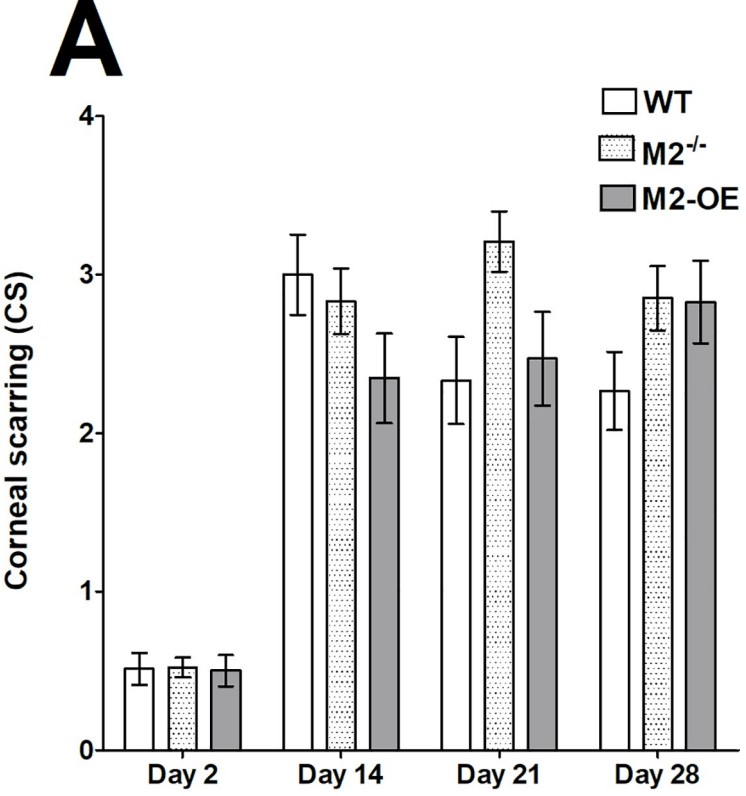

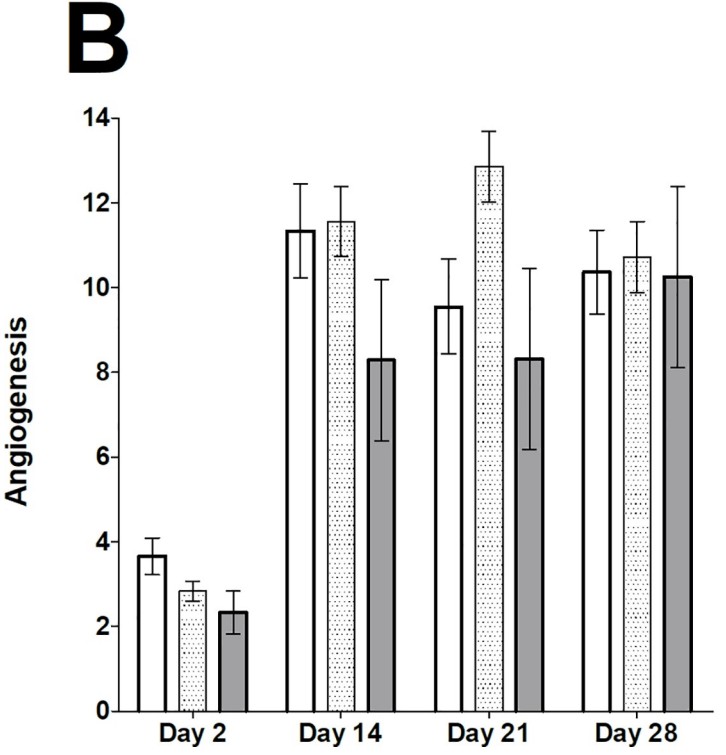

**Fig 7. Corneal scarring (CS) and angiogenesis of ocularly infected mice.** Mice were ocularly infected in both eyes with 2 X $10^5$ PFU/eye of McKrae virus. Corneal scarring (CS) and angiogenesis in surviving mice were examined on days 2, 14, 21, and 28 PI as described in Materials and Methods. The CS and angiogenesis scores represents the

average ± SEM from 40 eyes for each group of mice. p-values were determined using two-way ANOVA. Panels: A) CS; and B) Angiogenesis

Additionally, the normal time to reactivation in M2-OE mice could be due to increased IFN-γ expression, which has been shown to affect reactivation [44]. IFN-γ has also been shown to cause increased eye disease [45]. As we did not find a correlation between elevated IFN-γ levels and increase in eye disease in our M2-OE mice, it is possible that M2 anti-inflammatory role protects mice from eye disease.

Persistent and latent viral infections have been shown to lead to dysfunctional T cell responses and T cell exhaustion [46], characterized by expression of exhaustion markers such as PD1 and Tim3 [32]. IFN-γ expression is also known to upregulate PD1 [47]. We previously showed that increased latency in the TG of WT mice is associated with higher CD8, PD1, and Tim3 expression [32, 48]. However, we did not find a correlation between the absence or over-expression of M2 with T cell responses and exhaustion markers. Further, latency levels did not correlate with levels of CD8, PD1, and Tim3 expression suggesting that while the imbalance of M1/M2 contributes to increased phagocytosis, primary virus replication, and latency, it does not contribute to any specific patterns of T cell exhaustion. Our results underscore the importance of a tightly controlled balance of M1/M2 macrophages. Although M2 macrophages are thought to be more protective and play a major role in tissue repair as is demonstrated in a cystic fibrosis model of lung injury [49]. Our study clearly shows that macrophage balance is critical to control HSV-1 primary and latent infections. This finding is consistent with a previous study which found that disturbing the equilibrium between the M1/M2 functional axis can lead to chronic diseases [50]. M2 activation has also been associated with increased lung pathology in a SARS-CoV-1 mouse model [51]. It is possible that the increased number of M2 macrophages in M2-OE mice could suppress the ability of M1 macrophages to control viral replication. Alternatively, M2 macrophages in M2-OE mice may have re-polarized toward M1, as previous studies have reported that environmental cues can reverse the original phenotype [52]. Yet another study demonstrated that adipose tissue associated macrophages switch from an M2-like phenotype to a classically activated M1-like phenotype [53].

In summary, we show here that overexpression of M2 macrophages results in increased viral replication and viral load, increased levels of latency but not reactivation, increased expression of both pro- and anti-inflammatory cytokines but not increased T cell exhaustion. Taken together these results suggest a positive correlation between higher M2 expression and higher phagocytosis, while these two are negatively correlated with virus replication in the eye and higher latency. Our results further stress that a better understanding of the mechanistic pathways regulating macrophage polarization, macrophage trafficking, and their effects on other immune cells is needed to thoroughly understand their roles in HSV-1 infection as well as many other disease models. A barrier to the effective alleviation of the public health burden of HSV-1-induced corneal damage and blindness has been the identification and efficient targeting of the main immune responses. Our study suggest that a balance of M1 and M2 phenotypes will reduce ocular viral load and reduce corneal damage and latency-reactivation.

## Materials and methods

### Ethics statement

All animal procedures were performed in strict accordance with the Association for Research in Vision and Ophthalmology Statement for the Use of Animals in Ophthalmic and Vision Research and the NIH *Guide for the Care and Use of Laboratory Animals* (ISBN 0-309-05377-

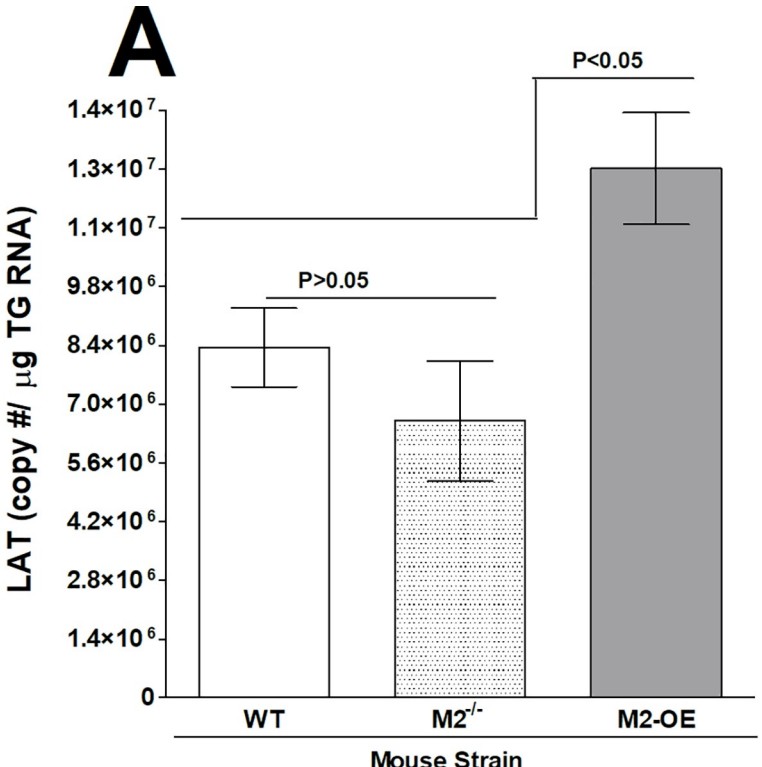

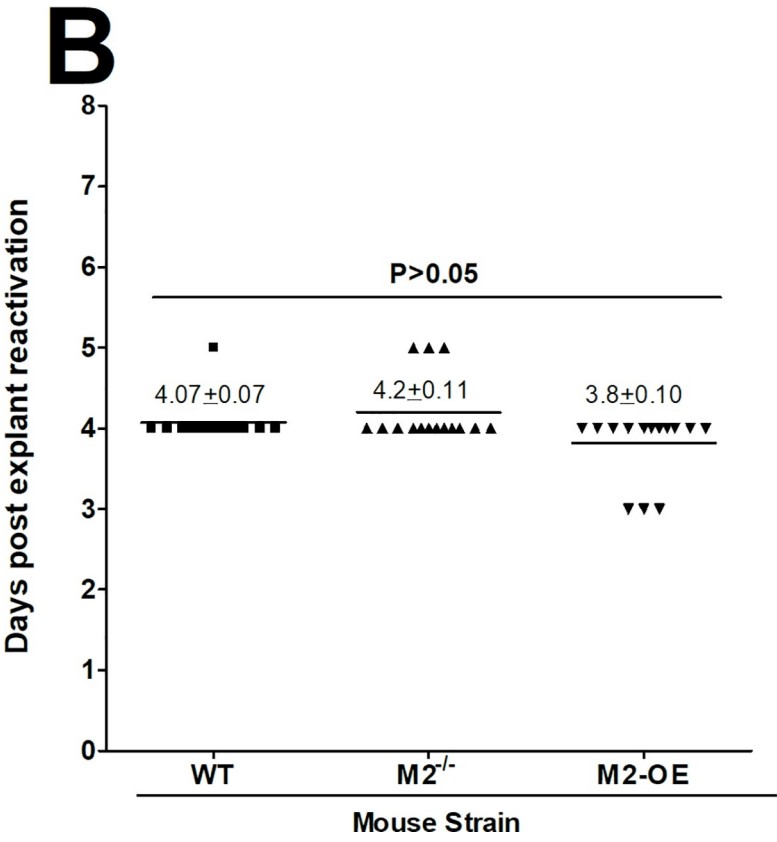

**Fig 8. Level of latency and effect of HSV-1 infection on induced reactivation kinetics in explanted TG from latently infected mice.** (A) Level of latency in TG from latently infected mice. WT, M2$^{-/-}$, and M2-OE mice were ocularly infected with 2 X 10$^5$ PFU/eye of McKrae virus. To analyze latency levels in the TG of latently infected mice on day 28 PI, TG were harvested and Quantitative RT-PCR was performed on the individual TG from each mouse. In each experiment, an estimated relative LAT copy number was calculated using standard curves generated from pGEM-5317. Briefly, DNA template was serially diluted 10-fold such that 5 μl contained from 10$^3$ to 10$^{11}$ copies of LAT then subjected to TaqMan PCR with the same set of primers. The copy number for each reaction was determined by comparing the normalized threshold cycle of each sample to the threshold cycle of the standard. GAPDH expression was used to normalize the relative expression of LAT RNA in the TG. Latency is based on 20 TG per each group of mice. p-values were determined using one-way ANOVA. (B) Explant reactivation in TG from latently infected mice. TG from latently infected mice were individually isolated on day 28 PI. Each individual TG was incubated in 1.5 ml of tissue culture media at 37°C. Media aliquots were removed from each culture daily for up to 5 days and plated on indicator cells (RS cells) to assess the appearance of reactivated virus. Results are plotted as the number of TG that reactivated daily. Numbers indicate the average time that the TG from each group first showed CPE ± SEM. Reactivation is based on 20 TG from ten mice per group. p-values were determined using one-way ANOVA. Panels: A) LAT copy number; and B) Reactivation.

3). Animal research protocols were approved by the Institutional Animal Care and Use Committee of Cedars-Sinai Medical Center (Protocol #5030).

## Mice

6-8-week-old C57BL/6 WT mice were purchased from The Jackson Laboratory (Bar Harbor, ME, USA). M2$^{-/-}$ mice (Previously called as GATA3 CKO) were generated by crossing GATA3 floxed (GATA3$^{fl/fl}$) mice (a gift from Dr. William E. Paul, and described in [54] with LysM-Cre mice (purchased from The Jackson Laboratory) to generate homozygous mGATA3KO mice [29]. M2-OE mice were generated by inserting a loxP-flanked STOP sequence following the promoter into the Rosa26 locus of *Rosa26$^{GATA3}$* mice (Gift from Dr. Maxime Bouchard, described in [55]). Rosa26 mouse line was used to overcome challenges associated with macrophage permeabilization and quantification. This reporter Rosa26$^{GATA3}$ mouse is generated by homologous recombination of the human *GATA3* (hGATA3) into the ubiquitously expressed *Rosa26* locus (designated as M2-OE hereafter). The expression of *GATA3* is activated by the Cre-mediated deletion of a stop cassette located upstream of the *GATA3* cDNA [52]. Therefore, only human GATA3, but not mouse GATA3, is expressed in the M2-OE mouse. Genotyping of M2-OE mice confirmed the specific expression of hGATA3 and GFP in myeloid cells, but not in cultured T cells.

These mice were bred with Cre mice and then crossed with M2$^{-/-}$ mice that contain LysM promoter driven Cre in myeloid cells to generate M2-OE mice. The genetic background of M2$^{-/-}$ mice and M2-OE mice were identical to C57BL/6 WT mice with the only exception that M2$^{-/-}$ mice had GATA3 gene knockout and M2-OE mice had an overexpression of GATA3 gene expression. To genotype these mice, three primers (EGFP2f, wP and mwP) were used to amplify a 575 bp PCR band from the mutation R26$^{GATA3}$ mice and a 250 bp PCR band from the WT R26$^{GATA3}$.

All mice were bred and maintained in the Cedars-Sinai Medical Center pathogen–free animal facility. Homozygous pups appeared to be healthy and were of normal size and body weight.

## Viruses and cells

Plaque-purified, virulent, WT herpes simplex virus 1 (HSV-1) strain McKrae was used in the study. Rabbit skin (RS) cells were used to prepare virus stocks, culture mouse tear swabs, and determine growth kinetics. RS cells were grown in Eagle's minimal essential medium supplemented with 5% fetal bovine serum (FBS).

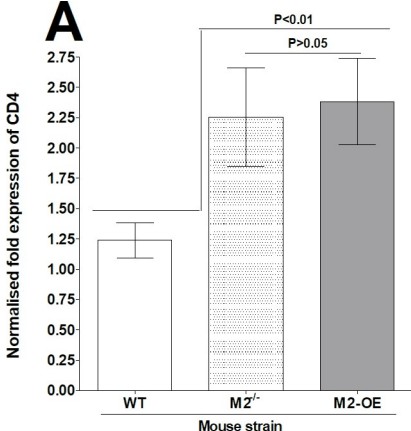
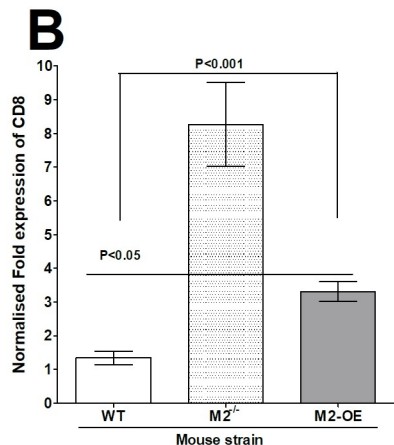

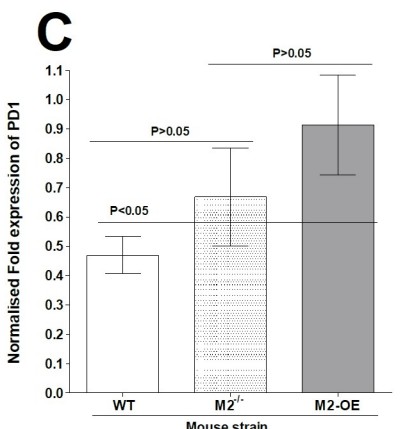
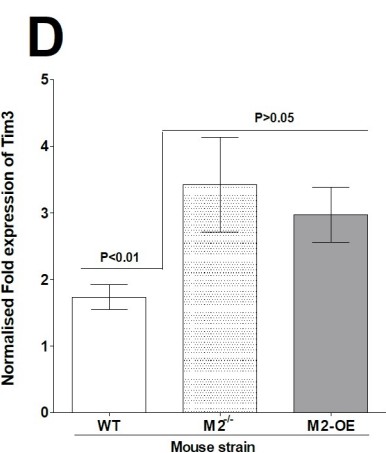

**Fig 9. Effect of HSV-1 infection on T cell exhaustion in TG of latently infected mice.** TG from latently infected WT, M2[-/-], and M2-OE mice were individually isolated on day 28 PI and qRT-PCR was performed using total RNA as described in the Materials and Methods. CD4, CD8, PD1, and Tim3 expression in naive mice for each group was used as a baseline to estimate relative expression of each transcript in TG of latently infected mice. GAPDH expression was used to normalize the relative expression of each transcript. Each point represents the mean ± SEM from 10 TG per group. Panels: A) CD4 transcript; B) CD8 transcript; C) PD1 transcript; and D) Tim3 transcript.

## Genotyping by PCR

For genotyping, tail samples were collected and lysed at 55°C overnight in 100 μl lysis buffer (100 mM Tris-HCl, pH 8.5, 5 mM EDTA, 0.2% SDS, 200 mM NaCl and 1 mg/ml of Proteinase K). After diluting 1:10 in distilled water, 1 μl was used as a PCR template. Primer sequences were as follows: 5′-TCAGGGCACTAAGGGT TGTTAACTT-3′ (P8); 5′-GAATTCCATC-CATGAGACACACAA-3′ (P11); 5′- CAGTCTCTGGTATTGATCTGCTTCTT-3′ (P13); 5′-GTGCAGCAGAGCAGGAAACTCTCAC-3′ (P16);5′-CCCAGA AATGCCAGATTACG-3' (P3066); 5′-CTTGGGCTGCCAGAATTTCTC-3' (P3067); and 5'-TTA CAGTCGGCCAG GCTGAC-3' (P3068).

## Ocular infection

Mice were infected with $2 \times 10^5$ PFU per eye of McKrae virus as an eye drop in 2 μl of tissue culture media as we described previously [19, 20]. Corneal scarification was not performed prior to infection.

## Preparation of macrophages from mouse peritoneal cavities

Peritoneal macrophages (PM) were harvested from WT, M2$^{-/-}$, and M2-OE mice after expansion by treating the mice with zymosan (Sigma-Aldrich, St. Louis, MO) as we described previously [19, 20]. Mice received 200 μl of 2-mg/ml zymosan in sterile, 0.9% (wt/vol) saline solution in one intraperitoneal injection. On the third day, the peritoneal membrane was separated from under the abdominal musculature. 5–7 ml ice cold phosphate-buffered saline (PBS) was injected into peritoneal cavity. The peritoneum was gently and completely massaged, and PBS was aspirated from peritoneal cavity. The peritoneal fluid was cultured overnight in complete Dulbecco modified Eagle medium (DMEM) with 10% FBS in 24-well plates. The medium was removed and the cells were washed twice with PBS to remove any floating cells. Adherent cells were recovered by gentle scraping and subjected to the M1 and M2 macrophage activation protocols described below.

## Activation and infection of PM *in vitro*

Cells were seeded at $2 \times 10^5$ cells per well in a 24-well plate. After overnight incubation, the medium was replaced with fresh complete DMEM containing 50 ng/ml of murine IFN-γ (Peprotech, Rocky Hill, NJ) and 100 ng/ml of lipopolysaccharide (LPS; Sigma-Aldrich, St. Louis, MO) for M1 activation, or complete DMEM containing 10 ng/ml of murine IL-4 (Peprotech) for M2 activation as we described previously [19, 20, 29]. On the following day, cells were infected with 10 PFU/cell of HSV-1 for 1 hr. Infected cells were then washed three times with PBS and fresh complete DMEM was added to each well. Infected cell monolayers were frozen at 12 and 24 hr PI. After two cycles of freeze thawing of infected cells, virus titer was determined by standard plaque assay using RS cells as described [56].

## Preparation of cells from bone marrow

Femoral bones were dissected and all the remaining tissue on the bones was removed. Each bone end was cut off and bone marrow was expelled. Bone marrow cells were cultured for 6 days. To differentiate and activate macrophages, 20ng/ml M-CSF or GM-CSF {Peprotech, Rocky Hill, NJ Catalog no. 315–02 (M-CSF) and 315–03 (GM-CSF)} was added along with the cells to be cultured as we described previously [30]. On day 3, M-CSF and GM-CSF were added again and cells were returned to the incubator until day 6. On day 6, petri plates were washed 3 times with PBS, removing all floating cells. Macrophages adhere to the petri plates and were scraped off and counted for further procedures.

## Phagocytosis assay

Bone marrow cells were cultured and differentiated into macrophages as described above. Phagocytosis assay was performed using a Phagocytosis Assay Kit (IgG-FITC) from Cayman chemical (Ann Arbor, Michigan) according to manufacturer instructions. After counting, cells were plated in 24-well plates. After overnight culture, some of the adhered cells were infected with HSV-1 or used as mock control. At 24 hr PI, latex beads-rabbit IgG-FITC complex (Item No. 400291) were added directly to the pre-warmed culture medium to a final dilution of 1:100 to 1:500 for two hr. After 2 hr, cells were washed with buffer supplied in the kit and

surface staining with F4/80 AF 594 antibody was performed to gate the total macrophages with no cell fixation. Cells were then transferred to FACS tubes for flow cytometry. Phagocytosis was measured using FACS DIVA software. No FITC signal was detected in untreated mock control BM cells in any group.

## Viral titers from tears of infected mice

Tear films were collected from 20 mouse eyes per group on days 1–7 PI using a Dacron-tipped swab [19, 20]. Each swab was placed in 1 ml of tissue culture medium and squeezed. The amount of virus was determined by a standard plaque assay on RS cells.

## Monitoring corneal scarring (CS) and angiogenesis

The severity of CS lesions in mouse corneas was examined by slit-lamp biomicroscopy. Scoring scale was: 0, normal cornea; 1, mild haze; 2, moderate opacity; 3, severe corneal opacity but iris visible; 4, opaque and corneal ulcer; 5, corneal rupture and necrotizing keratitis as we described [57]. The severity of angiogenesis was recorded using a system in which a grade of 4 for a given quadrant of the circle represents a centripetal growth of 1.5 mm toward the corneal center. The score of the four quadrants of the eye was summed to derive the neovessel index (range, 0–16) for each eye at a given time point [58]. Each cornea was examined and the mean ± SEM was calculated for each group.

## *In vitro* explant reactivation assay

Mice were sacrificed on day 28 PI and individual TG were removed and cultured in tissue culture media as described [19, 20, 59]. Aliquots of media were removed from each culture daily and plated on indicator cells (RS cells) to detect reactivated virus. As the media from explanted TG cultures was plated daily, we could determine the time at which reactivated virus first appeared in the explanted TG cultures.

## RNA extraction, cDNA Synthesis, and TaqMan RT-PCR

TG from individual mice were collected on day 28 PI, immersed in RNA later, RNA Stabilization Reagent (Thermo Fisher Scientific, Waltham, MA, USA), and stored at −80°C until processing. Total RNA was extracted as described [19, 20, 59]. Levels of LAT RNA from latent TG were determined using a custom-made LAT primers and probe as follows: forward primer, 5′-GGGTGGGCTCGTGTTACAG-3′; reverse primer, 5′-GGACGGGTAAGTAACAGAGTCTCTA-3′; and probe, 5′-FAM-ACACCAGCCCGTTCTTT-3′ (amplicon length = 81 bp).

Levels of CD4, CD8, PD1, and Tim3 transcripts in TG were evaluated using commercially available TaqMan Gene Expression Assays (Applied Biosystems, Foster City, CA, USA) with optimized primer and probe concentrations. Assays used in this study were as follows: (1) CD4, ABI Mm00442754_m1 (amplicon length = 72 bp); (2) CD8 (α chain), ABI Mm01182108_m1 (amplicon length = 67 bp); (3) PD1 (programmed death 1; also known as CD279), ABI Mm00435532_m1 (amplicon length = 65 bp); (4) Tim3 (ABI Mm00454540_m1; amplicon length, 98 bp) and (5) GAPDH used to normalize transcripts, ABI Mm99999915_G1 (amplicon length = 107 bp).

For corneal tissues and cultured cells, total RNA was extracted and 1ug of total RNA was reverse transcribed using a high-capacity cDNA reverse transcription kit (Applied biosystems, CA) according to manufacturer's protocol. mRNA expression levels of the genes in the study were determined using: (1) NOS2; assay ID (Thermo Fisher), Mm00440502_m1; amplicon size, 66 bp; (2) ARG1; Mm00475988_m1; 65 bp; (3) IFN-γ; Mm00801778_m1; 101 bp; (4) IL-

4, Mm00445259_m1 (79 bp). gB-specific primers were used to measure the viral transcript in corneas of infected mice on day 4 PI (forward, 5′-AACGCGACGCA CATCAAG-3′; reverse, 5′-CTGGTACGCGATCAGAAAGC-3′) and probe (5′-6-carboxyfluorescein {FAM}-CAGC CGCAGTACTACC-3′). The amplicon length for this primer set is 72 bp. Relative gB DNA copy numbers were calculated using standard curves generated from the plasmid pAc-gB1 [60]. By comparing the normalized threshold cycle ($C_T$) of each sample to the threshold cycle of the standard curve, the copy number for each reaction product was determined. For fold change of expression analysis, the $2^{-\Delta\Delta CT}$ method was used to calculate gene expression fold change compared to expression in uninfected controls.

## Statistical analysis

For all statistical tests, p-values less than or equal to 0.05 were considered statistically significant and marked by a single asterisk (*). P-values less than or equal to 0.001 were marked by double asterisks (**). A two-tailed student t-test with unequal variances was used to compare differences between two experimental groups. A one-way ANOVA test was used to compare differences among three or more experimental groups. All experiments were repeated at least three times to ensure accuracy.

## Author Contributions

**Conceptualization:** Homayon Ghiasi.

**Data curation:** Ujjaldeep Jaggi, Mingjie Yang, Harry H. Matundan, Satoshi Hirose.

**Formal analysis:** Ujjaldeep Jaggi, Homayon Ghiasi.

**Funding acquisition:** Homayon Ghiasi.

**Investigation:** Ujjaldeep Jaggi, Prediman K. Shah, Homayon Ghiasi.

**Methodology:** Ujjaldeep Jaggi, Mingjie Yang.

**Project administration:** Homayon Ghiasi.

**Resources:** Prediman K. Shah, Homayon Ghiasi.

**Software:** Ujjaldeep Jaggi, Homayon Ghiasi.

**Supervision:** Homayon Ghiasi.

**Validation:** Ujjaldeep Jaggi, Homayon Ghiasi.

**Visualization:** Ujjaldeep Jaggi, Homayon Ghiasi.

**Writing – original draft:** Ujjaldeep Jaggi, Homayon Ghiasi.

**Writing – review & editing:** Ujjaldeep Jaggi, Behrooz G. Sharifi, Homayon Ghiasi.

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
