## [Decision Letter · Decision Letter 0]

23 Aug 2020

August 21st, 2020

Dear Dr. Ghiasi,

Thank you for submitting your manuscript “Increased phagocytosis in the presence of enhanced M2-like macrophage responses correlates with increased primary and latent HSV-1 infection” for review by PLOS Pathogens. As with all papers submitted to the journal, yours was evaluated by a PLOS Pathogens Associate Editor in consultation with the Editorial Board. Your article was also evaluated by three independent reviewers. All three reviewers were enthusiastic about the studies and recommended minor revisions. We therefore ask you to modify the manuscript according to the review recommendations before we can consider your manuscript for acceptance. Your revisions should address the specific points made by each reviewer.

1. There is no discussion of so-called “M0”macrophages. Do they exist in the M2-/- and M2- OE mice? If so, to what extent and to what end pathogenetically?

2. Peritoneal macrophages are collected from wildtype, M2-/-, and M2-OE mice for initial comparisons. While the findings are convincing and compelling, do the peritoneal macrophage populations faithfully represent the macrophage populations of the cornea during primary HSV-1 infection?

3. Some clarification is needed regarding the precise role of macrophages during primary HSV-1 infection of the corneal surface. It is understood that (1) macrophages are recruited to the corneal surface during primary HSV-1 infection, and (2) macrophages function has phagocytes during primary HSV-1 infection. The authors then conduct an experiment to compare peritoneal macrophages from wildtype mice, M2-/- mice, and M2-OE mice for HSV-1 replication. Why? Do the authors believe that the infiltrating macrophages, in addition to serving as phagocytes, also serve as a target for virus infection and replication and thereby increase the amount of virus found in tear films? If so, what percentage of infiltrating macrophages become infected during the course of primary HSV-1 infection?

4. Under introduction please add a couple of sentences on HVS-1 ocular pathogenesis. Please also comment whether HSV-1 is known to cause primary infection in the cornea.

5. Line 104, please explain the functional need for Rosa26.\\

6. Line 212, please cite a reference for exhaustion markers used in the study.

7. Line 336, please add the original source for viral strains used in the study.

8. Fig. 1, please expand HE and HO in the figure legend.

9. In methods, please provide details of how normalized fold expression was calculated.

10. In Figure 6, did authors measure the protein level of IL-4 and IFN-g in the corneas of infected mice? Please comment.

11. In Figure 9, the author showed higher level of PD-1 in TG of M2 over expressing mice. What cell population is expressing higher level of PD-1 in the latent TG?

12. In the discussion, can authors discuss the translational implications of their result.

Please prepare and submit your revised manuscript within 30 days. If you anticipate any delay, please let us know.

Important additional instructions are given below your reviewer comments. Please note while forming your response, if your article is accepted, you may have the opportunity to make the peer review history publicly available. The record will include editor decision letters (with reviews) and your responses to reviewer comments. If eligible, we will contact you to opt in or out.

Sincerely,

Clinton Jones, PhD

Guest Editor

PLOS Pathogens

Erik Flemington

Section Editor

PLOS Pathogens

Kasturi Haldar

Editor-in-Chief

PLOS Pathogens

orcid.org/0000-0001-5065-158X

Michael Malim

Editor-in-Chief

PLOS Pathogens

orcid.org/0000-0002-7699-2064

Reviewer Comments (if any, and for reference):

Reviewer's Responses to Questions

**Part I - Summary**

Reviewer #1: Summary: Previous work from the Ghiasi laboratory has shown that macrophages of the M2 phenotype are more protective than macrophages of the M1 phenotype against HSV-1 primary corneal infection and latency. Ghiasi and coworkers continued their investigations of M1 vs M2 macrophage phenotypes on the pathogenesis of HSV-1 corneal disease in the present investigation through development and use of novel C57BL/6 mice that either lack M2 macrophages (M2-/- mice) or overexpress M2 macrophages (M2-OE). This investigation examined further the impact of a deficiency in M2 macrophages as well as the impact of overexpression of M2 macrophages on HSV-1 corneal disease and latency at the ganglionic level with some emphasis on the phagocytotic activity of macrophages that infiltrate the cornea following primary HSV-1 corneal infection. Analysis of several experimental endpoints appropriate for a study of HSV-1 corneal infection and latency revealed no effect on phagocytosis, virus replication as measured in tear films for infectious virus, ganglionic latency, and cytokine expression when M2-/- mice were compared with wildtype mice. In contrast, overexpression of M2 macrophages in M2-OE mice when compared with wildtype mice revealed increased phagocytosis, increased virus replication at the corneal surface during primary infection, increase ganglionic latency, and increased expression of cytokines, both pro-inflammatory and anti-inflammatory. The authors conclude that the proper balance of M1 vs M2 macrophage phenotypes is important in controlling HSV-1 corneal disease.

Review: This is a well written manuscript that summarizes the findings of an extension of work previously published by the Ghiasi laboratory on the role of M1 vs M2 macrophages during the pathogenesis of primary HSV-1 corneal infection, ganglionic latency, and ganglionic reactivation. Little is known about macrophage phenotypes and HSV-1 disease pathogenesis, so this investigation is a worthwhile pursuit. Of greater significance, the study uses two mouse strains unique to the Ghiasi laboratory. The studies are carried out carefully with appropriate use of statistical analysis. While no major concerns are apparent, attention to a few minor issues would serve to strengthen the manuscript.

1. There is no discussion of so-called “M0” macrophages. Do they exist in the M2-/- and M2-OE mice? If so, to what extent and to what end pathogenetically?

2. Peritoneal macrophages are collected from wildtype, M2-/-, and M2-OE mice for initial comparisons. While the findings are convincing and compelling, do the peritoneal macrophage populations faithfully represent the macrophage populations of the cornea during primary HSV-1 infection?

3. Some clarification is needed regarding the precise role of macrophages during primary HSV-1 infection of the corneal surface. It is understood that (1) macrophages are recruited to the corneal surface during primary HSV-1 infection, and (2) macrophages function has phagocytes during primary HSV-1 infection. The authors then conduct an experiment to compare peritoneal macrophages from wildtype mice, M2-/- mice, and M2-OE mice for HSV-1 replication. Why? Do the authors believe that the infiltrating macrophages, in addition to serving as phagocytes, also serve as a target for virus infection and replication and thereby increase the amount of virus found in tear films? If so, what percentage of infiltrating macrophages become infected during the course of primary HSV-1 infection?

Reviewer #2: This is an interesting manuscript that focuses on describing the significance of M1/M2 macrophages in HSV-1 primary and latent infections. Until now very little has been known on their specific roles in HSV-1 infection. To address the significance the authors generated M2-/-mice as well as conditional transgenic mice overexpressing M2 macrophages. They found clear evidence that higher expression of M2 macrophages correlated with higher phagocytosis, as well as higher HSV-1 replication. The latter led to an increase in latency. At the cytokine level they also noticed an increase in IL-4 and IFN-γ expression. Quite interestingly, they found that the lack of M2 macrophages did not affect HSV-1 infectivity. Manuscript is well written, provides highly significant information and easy to follow. The main conclusion that a balanced proportion of M1/M2 macrophage may be an effective way to control the infection is well supported by the results shown in the manuscript.

Reviewer #3: This is an excellent study from Dr. Ghiasi’s group. In this study, the authors showed that over expression but not lack of M2 macrophage population increases the load of HSV-1 replication and latency in infected mice. However, over expression of M2 population does not affect HSV-1 reactivation. The conclusions made were strongly supported by the data presented in the manuscript. Strength of the study involve the use of M2 knockout and M2 over expressing mice in studying the role of M2 macrophage population in HSV-1 infectivity, latency and corneal disease. Overall, a well thought study delineating the role of M2 macrophage population in the mouse model of corneal HSV-1 infection.

**Part II – Major Issues: Key Experiments Required for Acceptance**

Reviewer #1: No additional key experiments are required.

Reviewer #2: None.

Reviewer #3: No new experiments are needed to support the conclusions drawn from the data presented.

**Part III – Minor Issues: Editorial and Data Presentation Modifications**

Reviewer #1: The subheading within the Results section of Lines 177 and 176 is awkward and needs revision.

Reviewer #2: Under introduction please add a couple of sentences on HVS-1 ocular pathogenesis. Please also comment whether HSV-1 is known to cause primary infection in the cornea.

Line 104, please explain the functional need for Rosa26.

Line 212, please cite a reference for exhaustion markers used in the study.

Line 336, please add the original source for viral strains used in the study.

Fig. 1, please expand HE and HO in the figure legend.

Fig.2, this and other figures please uniformly use either WT (listed in the legend) or control (listed in the figure).

Reviewer #3: 1. In methods, please provide details of how normalized fold expression was calculated.

2. In Figure 6, did authors measure the protein level of IL-4 and IFN-g in the corneas of infected mice? Please comment.

3. In Figure 9, the author showed higher level of PD-1 in TG of M2 over expressing mice. What cell population is expressing higher level of PD-1 in the latent TG?

4. In the discussion, can authors discuss the translational implications of their results

PLOS authors have the option to publish the peer review history of their article (what does this mean?). If published, this will include your full peer review and any attached files.

Reviewer #1: No

Reviewer #2: No

Reviewer #3: No
---

## [Editor Report · Decision Letter 1]

9 Sep 2020

Dear Dr. Ghiasi,

We are pleased to inform you that your manuscript 'Increased phagocytosis in the presence of enhanced M2-like macrophage responses correlates with increased primary and latent HSV-1 infection' has been provisionally accepted for publication in PLOS Pathogens.

Best regards,

Clinton Jones, PhD

Guest Editor

PLOS Pathogens

Erik Flemington

Section Editor

PLOS Pathogens

Kasturi Haldar

Editor-in-Chief

PLOS Pathogens

orcid.org/0000-0001-5065-158X

Michael Malim

Editor-in-Chief

PLOS Pathogens

orcid.org/0000-0002-7699-2064

The authors have satisfactorily addressed the minor comments of the three reviewers.
---

## [Editor Report · Acceptance letter]

30 Sep 2020

Dear Dr. Ghiasi,

We are delighted to inform you that your manuscript, "Increased phagocytosis in the presence of enhanced M2-like macrophage responses correlates with increased primary and latent HSV-1 infection," has been formally accepted for publication in PLOS Pathogens.

Best regards,

Kasturi Haldar

Editor-in-Chief

PLOS Pathogens

orcid.org/0000-0001-5065-158X

Michael Malim

Editor-in-Chief

PLOS Pathogens

orcid.org/0000-0002-7699-2064